# COVID-19 and Sickle Cell Disease in the Province of Quebec, Canada: Outcomes after Two Years of the Pandemic

**DOI:** 10.3390/jcm11247361

**Published:** 2022-12-12

**Authors:** Mathias Castonguay, Nawar Dakhallah, Justin Desroches, Marie-Laure Colaiacovo, Camille Jimenez-Cortes, Anne-Marie Claveau, Samuel Bérubé, Amer Yassine Hafsaoui, Amalia Souza, Pauline Tibout, Christophe Ah-Yan, Anne-Marie Vincent, Veronique Naessens, Josée Brossard, Sharon Abish, Raoul Santiago, Denis Soulières, Vincent Laroche, Yves Pastore, Thai Hoa Tran, Stéphanie Forté

**Affiliations:** 1Division of Hematology and Oncology, Department of Medicine, Centre Hospitalier de l’Université de Montréal, Montreal, QC H2X 3E4, Canada; 2Division of Pediatric Hematology and Oncology, Department of Paediatrics, Centre Hospitalier Universitaire de Sainte-Justine, Montreal, QC H3T 1C5, Canada; 3Division of Hematology and Oncology, Department of Medicine, McGill University Health Center, Montreal, QC H4A 3J1, Canada; 4Division of Hematology and Oncology, Department of Medicine, Maisonneuve-Rosemont Hospital, Montreal, QC H1T 2M4, Canada; 5Division of Pediatric Hematology and Oncology, Department of Paediatrics, Centre Hospitalier de l’Université Laval, Quebec, QC G1V 4G2, Canada; 6Division of Pediatric Hemato-Oncology, Centre Hospitalier Universitaire de Sherbrooke, Sherbrooke, QC J1H 5H3, Canada; 7Division of Pediatric Hematology and Oncology, Department of Paediatrics, Montreal Children Hospital, Montreal, QC H4A 3J1, Canada; 8Division of Hematology and Oncology, Department of Medicine, Centre Hospitalier de l’Université Laval, Quebec, QC G1V 4G2, Canada

**Keywords:** sickle cell disease, COVID-19, acute chest syndrome, viral infection, Quebec, Canada

## Abstract

Background: Patients with sickle cell disease (SCD) are considered at higher risk of severe COVID-19 infection. However, morbidity and mortality rates are variable among countries. To date, there are no published reports that document outcomes of SCD patients with COVID-19 in Canada. Methods: A web-based registry was implemented in June 2020 capturing outcomes of SCD patients with COVID-19 from March 2020 to April 2022 and comparing them to the general population of Quebec, Canada. Results: After 24 months of the pandemic, 185 SCD patients with confirmed SARS-CoV-2 infection were included in the registry. Overall, the population was young (median age 12 years old) and had few comorbidities. No deaths were reported. Risk of hospitalization and admission to intensive care unit (ICU) because of COVID-19 was higher in patients with SCD than in the general population (relative risks (RR) 5.15 (95% confidence interval (95% CI) 3.84–6.91), *p* ˂ 0.001 and 4.56 (95% CI 2.09–9.93) *p* ˂ 0.001). A history of arterial hypertension or acute chest syndrome in the past 12 months was associated with a higher risk of severe disease (RR = 3.06 (95% CI 1.85–5.06) *p* = 0.008 and 2.27 (95% CI 1.35–3.83) *p* = 0.01). Hospitalized patients had lower hemoglobin F than non-hospitalized patients (12% vs. 17%, *p* = 0.02). For those who had access to vaccination at the time of infection, 25 out of 26 patients were adequately vaccinated and had mild disease. Conclusions: The SCD population is at higher risk of severe disease than the general population. However, we report favorable outcomes as no deaths occurred. Registries will continue to be critical to document the impact of novel COVID-19 specific therapy and vaccines for the SCD population.

## 1. Introduction

On 11 March 2020, the World Health Organization (WHO) declared the severe acute respiratory syndrome coronavirus 2 (SARS-CoV-2) outbreak a global pandemic [1]. Many concerns were raised regarding higher complication rates of this virus for patients suffering from preexisting medical conditions, such as sickle cell disease (SCD).

SCD is the most common inherited hemoglobinopathy worldwide [2]. The most prevalent genotype is homozygous HbSS, followed by compound heterozygous HbS/β^0^-thalassemia, HbSC, and HbS/β^+^-thalassemia. SCD presents with acute exacerbations of chronic hemolysis and acute complications such as episodic vaso-occlusive crises (VOC), acute chest syndrome (ACS) and strokes. Cardiac dysfunction, pulmonary hypertension, nephropathy and splenic dysfunction are other chronic complications. SCD is also associated with a procoagulant state with a heightened risk of thromboembolic events [3]. For all these reasons, patients with SCD are considered at high risk of severe COVID-19 infection.

Viral respiratory infections are known triggers of ACS, one of the major causes of mortality in patients with SCD [4,5,6]. For example, during the H1N1 pandemic in 2009, high rates of complications among SCD patients were reported. These included higher rates of pain crisis, ACS (up to 34% of patients), hospitalization, intensive care unit (ICU) admission, and mechanical ventilation requirement [7,8]. Severe disease was more common in older patients with prior ACS [8].

To this day, the impact of COVID-19 in the SCD population remains uncertain. Concerns were raised at the beginning of the pandemic as Panepinto and colleagues reported hospitalization and mortality rates as high as 67 and 7% respectively during the first wave (March–May 2020) [9]. However, several recent studies have demonstrated more favorable outcomes for SCD patients [10,11,12,13].

The aim of this SCD registry study Is to report the epidemiological and clinical characteristics of SCD patients diagnosed with SARS-CoV-2 infection between March 2020 and April 2022, and to compare the hospitalization and ICU admission rates, as well as mortality rates of those patients to Quebec’s general population using epidemiological data from the National Institute of Public Health of Quebec (INSPQ) [14,15]. We hypothesize that SCD patients are at higher risk of severe SARS-CoV-2 infection, and that SCD-associated preexisting health conditions worsen outcomes.

## 2. Materials and Methods

This study was approved by the ethics committee of the Centre Hospitalier Universitaire Saint-Justine and participating centers. Research was conducted in accordance with the principles of Helsinki.

### 2.1. Study Design

A multicentric web-based SCD-COVID-19 registry to capture SARS-CoV-2 infections among SCD patients in the province of Quebec was implemented in June 2020. Seven centers, following the majority of SCD patients in Quebec, contributed to the registry (4 pediatric and 3 adult hospitals).

Data collection was done retrospectively for each wave using a REDCap standardized questionnaire.

### 2.2. Inclusion and Exclusion Criteria 

SCD patients of any age and genotype with a SARS-CoV-2 infection confirmed by polymerase chain reaction (PCR) at any of the 7 participating centers were included. Since January 2022, patients with a positive antigen test were also included. This corresponds to a period when access to PCR testing was restricted, and rapid antigen tests became broadly available.

Members of the participating SCD comprehensive centers (physicians, nurses, trainees) identified patients with mild to severe COVID-19. Charts of patients consulting and/or being admitted for COVID-19 infection were reviewed. Patients self-reporting COVID-19 and not requiring an ER visit or hospitalization were also included if they had undergone SARS-CoV-2 testing.

Cases between March 2020 and March 2022 were included. Screening for patients and data collection was performed continuously from July 2020 to April 2022.

### 2.3. Patient Characteristics 

Demographic information (age, sex), genotype, ABO blood group, SCD treatment, and medical comorbidities (including prior medical history of ACS, hypertension, pulmonary hypertension, chronic kidney failure, obesity) at the time of infection were collected. The number of emergency department (ED) visits for sickle cell related complications (mainly VOC), total hemoglobin (Hb, g/L) and the hemoglobin F fraction (Hb F) on electrophoresis as well as history of ACS up to 12 months prior to the infection were recorded. Information about vaccination status against SARS-CoV-2 was available for adults at the time of data cut-off. Since approval of anti-SARS-CoV-2 vaccines in the pediatric population was granted later, information about vaccination in those <18 was not included in this current analysis.

### 2.4. Outcome

The primary outcomes were defined as COVID-19-related hospitalization, admission to the ICU, and death of any cause at 30 days from the onset of the SARS-CoV-2 infection.

Secondary outcomes were the inaugural symptoms of COVID-19, severity of COVID-19 infection, complications, and treatments. Inaugural symptoms were classified as classic COVID-19 symptoms (cough, shortness of breath, fever, fatigue, myalgias, rhinorrhea, nasal congestion, anomia and ageusia) or SCD-related complications. Acute chest syndrome (ACS) was defined by a new pulmonary infiltrate associated with at least one of the following: fever, chest pain, hypoxia, tachypnea, wheezing or cough [16]. Severity of COVID-19 infection was classified as mild (not requiring hospitalization), moderate (hospitalization without ICU admission), severe (hospitalization with ICU admission) and death. Complications and treatments were extracted when reported by the medical care team in the clinical documentation.

### 2.5. Statistical Analyses

Descriptive statistics were used to present the patient characteristics at the time of SARS-CoV-2 infection. Summary statistics were computed as median [interquartile range] for continuous variables and frequencies for discrete variables.

Univariate analysis of potential predictors of hospitalization was done. Relative risks for hospitalization were calculated for each potential predictor and 95% confidence intervals were computed. Student t-test and chi-square tests were performed, as appropriate.

The incidence of COVID-19 infection and the rate of hospital admission in the Province of Quebec for the population under 70 years of age were calculated for the first four waves. During the fifth wave, due to COVID-19 shortage in testing centers, COVID-19 incidence reported by the INSPQ was limited to specific populations (health care workers, patients admitted to hospitals, and members of the First Nations) and the INSPQ did not report infection confirmed by rapid antigen testing (available since December 2021). Therefore, comparison with the general population was not possible with the Omicron and BA.2 variants starting from December 2021.

All tests were two-sided and a *p*-value < 0.05 was considered significant. The Pearson bivariate statistic was used to analyze the strength of correlations. IBM SPSS Version 26 was used.

## 3. Results

After 24 months of the pandemic, 185 SCD patients were infected with SARS-CoV-2 in the province of Quebec, Canada.

### 3.1. Baseline Characteristics of SCD Patients in Quebec Infected by COVID-19

Table 1 shows the baseline characteristics of our cohort. The median age was 12 years old (IQR: 5–25 years old) with age ranging from one month to 68 years old. Most of the cohort was under the age of 40, 64% of the patients were younger than 20, 28% were between 20 and 40, and 8.1% were aged between 40 and 70 years old. Sex distribution was 51% males and 49% females. SCD genotype distribution was: 65% SS or Sβ^0^ and 28% of SC genotype. Most of our patients were on disease-modifying therapy. Overall, 65% of the entire cohort was on hydroxyurea and 14% were on chronic transfusion programs. Our cohort had few comorbidities: 7.4% had an ACS in the past year and 12% were obese. Other comorbidities affected less than 5% of the cohort. Six percent of our population presented to the ED 4 times or more in the last year for SCD-related complications. The median Hb F was 15% and was reported for patients older than one year old, not on chronic exchange transfusion.

### 3.2. Clinical Presentation and Outcomes of Infection

Table 2 describes the clinical presentation and outcomes of SCD patients infected with SARS-CoV-2. Symptoms of infection were diverse. Sixteen percent of the patients were asymptomatic and were diagnosed based on screening policies (prior to a procedure, surgery, or hospital admission) or with a positive COVID-19 contact. For symptomatic patients, fever and cough were the most common symptoms. Pain crisis was present in 16% of patients at diagnosis. Overall, 28% of the cohort were hospitalized due to COVID-19 and 3.7% required ICU interventions. No deaths were reported.

Table 3 focuses on hospitalized patients and documents complications and treatments received. ACS was the most common complication of COVID-19 infection, affecting 43% of hospitalized patients. Other complications accounted for 5% or less. Non-invasive oxygen support and mechanical ventilation were required in 20% and 12% of the study population. All patients received thromboprophylaxis or therapeutic anticoagulation if there was evidence of thrombosis at diagnosis. No patient under thromboprophylaxis had a thrombotic event and no patient developed VOC following admission. Importantly, no deaths occurred in our cohort after 24 months of the pandemic.

Table 4 identifies risk factors for hospitalization in the SCD population. Among the preexisting health conditions, hypertension and previous ACS in the prior year were associated with a higher risk of hospitalization. Pain crisis as a presentation of infection is associated with an increased risk of hospitalization. Hospitalized patients had lower median fetal hemoglobin (Hb F), but no difference in Hb F was observed among hospitalized patients with or without ACS (8.6 vs. 14.4%, *p* = 0.22, not shown in table). Age, sex, genotype, ABO blood group, baseline SCD treatment, and number of visits to the ED in the prior year had no impact on the risk of hospitalization. Regarding genotypes, SC was present in 27% and 28% (0.98 [0.57–1.63], *p* = 0.90) and SS in 69% and 62% (1.27 [0.76–2.11], *p* = 0.34) of hospitalized and non-hospitalized patients respectively and was not associated with an increased risk of severe infection.

Once vaccination for adults was available (1 March 2021), 25 out of 26 patients eligible for vaccination were adequately vaccinated and all of them had a mild infection not requiring hospitalization. One pregnant patient was not vaccinated and was hospitalized without ICU admission. She received monoclonal antibodies directed against SARS-CoV-2 S protein.

### 3.3. Comparing SCD Population to Quebec’s General Population

Table 5 compares rates of infection, hospitalization, and ICU admission between SCD and the general population of Quebec. This analysis excludes the 5th and 6th waves of the pandemic (variants Omicron and BA.2) from December 2022, because prevalence of infection of the general population was no longer documented in Quebec (SCD patients in waves one to four *n* = 103). The incidence, hospitalization rates, and ICU admissions were significantly higher among SCD patients compared to the general population of Quebec.

## 4. Discussion

This is the first study presenting outcomes of SCD patients with COVID-19 infection in Canada. We report a generally favorable clinical course in our cohort, with no deaths observed and identified certain risk factors for a more severe disease course.

### 4.1. Interpretation

Despite a possible over-representation of more severe cases of COVID-19 in our SCD registry, we describe an overall favorable disease course. This occurred in the context of intense SARS-CoV-2 transmission. In fact, Quebec had the highest number of confirmed COVID-19 cases in Canada in the first wave of the pandemic [17]. Mortality was high, especially in the elderly living in nursing homes [18]. 

The favorable disease course contrasts with the French, American, and SECURE-SCD registries. Table 6 presents an overall comparison of our cohort to the other registries. Arlet et al. published an updated report of 319 hospitalized patients with SCD in France with a median age of 26 years (in which the pediatric population represented 27% of the cohort) and reported a mortality rate of 3% for adults and 2.2% for the overall population [10]. From January 20th to September 20th, outcomes of patients with COVID-19 infection with SCD or sickle cell trait (SCT) were compared to those of Black patients without SCD/SCT, using data retrieved from the American TriNetX research network. This study included 312 SCD and 449 SCT patients. No difference was found between patients with SCT and Black patients without SCD/SCT. However, a higher proportion of SCD patients had pneumonia or pain crisis compared with the matched Black population. Moreover, significantly more SCD patients were hospitalized (41.3% vs. 12.2%) and had pneumonia/ACS (23.1% vs. 9.6%). The rates of acute respiratory distress syndrome and mechanical ventilation were not significantly different between both groups (11.2% vs. 8.9%) [11]. The SECURE-SCD Registry is an international collaboration that included 750 patients, of whom 364 were children. Updated in March 2022, this registry reported hospitalization rates as high as 48%, ICU admission rates of 7%, and a need for mechanical ventilation of 3% [12]. Hospitalization was associated with prior high frequency of medical visits for pain in adults and children and SCD-related heart, lung or renal comorbidities in children [13]. The death rates of SCD patients in the French, American, and SECURE-SCD studies were 2.2% (no deaths under 20 years of age), 3.2% and 4.7% for adults (0.3% for children), respectively.

The younger age of our population and fewer baseline comorbidities compared to other registries may explain the better outcomes observed. Similar observations were made by the Greek COVID-19 and hemoglobinopathies registry [19,20]. Universal health care access may also partially explain this observation, as studies have demonstrated that healthcare privatization and underfunding are associated with excess COVID-19 mortality [21] but cannot fully explain results since the French health care system also possesses universal health care access and reports a higher mortality rate. We also hypothesize that regular follow-ups in specialized SCD centers for all patients in Quebec may help prevent severe disease. Further studies are needed to confirm the two latter hypotheses before any conclusion can be made.

Unlike the French and SECURE-SCD registries, we found no increased risk of severe infection when comparing SS or SC genotypes. The overall prevalence of SC genotype was higher compared to other registries (Table 6). However, it was not associated with an increased risk of hospitalization. This high proportion of SC patients in our cohort reflects the similarly high proportion of SC patients (31%) in Canada [22]. Unlike the SECURE-SCD registry, frequent ED visits (mostly for pain crisis) were not associated with an increased risk of severe infection [12,13]. Just as reported by the French registry, a history of previous ACS was associated with an increased risk of hospitalization in our population, and it is coherent with previous observations from the H1N1 pandemic [7,8].

VOC as a presenting symptom of COVID-19 was associated with a higher need of hospitalization for fluid resuscitation and pain management. Median total Hb and Hb F were lower among hospitalized patients compared to non-hospitalized patients (excluding children less than one year old and patients on transfusion exchange programs). We therefore hypothesize that higher Hb F levels may prevent COVID-19 related VOC. Among hospitalized patients, a non-statistical tendency toward lower Hb F was observed in patients who had COVID-19 related ACS compared to patients without ACS. Higher Hb F levels may prevent COVID-19 related ACS, but larger studies are necessary to confirm this observation. This observation is consistent with our current knowledge about Hb F: higher levels of Hb F (at baseline or due to hydroxyurea) reduces polymerization of deoxy sickle hemoglobin and decreases the risks of VOC and ACS [23].

Compared to the general population of Quebec, hospitalization and ICU admission rates were higher when analyzed by age group, but overall, they were similar to other SCD registries, meaning that the SCD population should be considered at risk for more severe infection. We hypothesize that higher infection rates may be explained by frequent contact with health care centers, a tendency of over-representation of Afro-American workers in hospitals and potentially by strict symptoms surveillance and rapid referral to ED emphasized by our systemic follow-up teams. 

In our population, specific COVID-19 treatments like Dexamethasone or monoclonal antibodies directed against interleukin-6 (IL-6) or against SARS-CoV-2 Spike protein were not commonly used, mainly because severe infections were observed preferentially before these therapies were available. Several experts have also recommended thromboprophylaxis for SCD patients hospitalized with COVID-19. The absence of thrombotic complications in our cohort might be explained by the adherence of such recommendations in all our patients. Whether COVID-19 increases or not the thromboembolic risk for SCD patients remains unknown. Indeed, a recent study found no increased risk of VTE when comparing SCD patients hospitalized with COVID-19 to SCD patients hospitalized for another reason [24].

We believe registries are critical to document the impact of the pandemic and inform rapid rational modifications to public health policies. With this registry, the Ministry of Health and Social Services (MSSS) of Quebec implemented a preventive withdrawal from work after the first wave and prioritized vaccination for the SCD population. A high proportion of patients (96%) were subsequently vaccinated, and they all had a mild infection. As several specific COVID-19 therapies are available now, registries will have a major role in documenting how the SCD population may or may not benefit from these therapies. 

### 4.2. Limitations

Our study has several limitations. First, the precise number of individuals living with SCD in the province of Quebec is unknown. The population is estimated to be approximately 1500, representing less than 0.02% of the total population of Quebec (8.5 million) [12]. This number was approximated from a survey of all comprehensive SCD centers in the Province of Quebec, which was conducted by HémaQuébec in 2017 (personal communication, Nancy Robitaille). Since 2016, a universal postnatal SCD screening program has been implemented in Quebec, and all children have been referred to comprehensive centers. Based on these approximations, it is conceivable that certain individuals with less severe SCD phenotypes are not actively followed in comprehensive SCD centers, and as a result their COVID-19 episodes would not be captured in the registry. This could introduce a bias towards generally describing more severe disease courses of COVID-19 in SCD. A second limitation of our study is that it was based on voluntary contributions to the registry by SCD experts. As a result, several COVID-19 episodes could have been omitted, especially as the pandemic progressed into its second year. Attempts were made to capture all COVID-19 episodes in SCD patients through the provincial program of COVID-19 testing (INSPQ). However, SCD was not considered a risk factor at the time the testing program was implemented, and patients did not systematically self-report their SCD diagnoses.

Along these lines, the retrospective nature of the study implies that some patients experiencing few or no symptoms may have been missed. This is particularly true after home antigen testing became widely available in the fall of 2021, since reported cases were dependent on self-declaration or on whether it was questioned by the systematic follow-up team. This selection bias could underestimate the current infection prevalence observed, which is already higher than the general population. Overall, these biases would have led to an over-representation of more severe cases in our study.

Finally, risk factor definitions, screening, and reporting may have varied between centers. For example, pulmonary hypertension diagnosis was not systematically reported as there is no provincial screening program using cardiac echocardiography for routine surveillance. These limitations in terms of potential clinical risk factor definitions may explain differences between our studies and others. However, more objective risk factors, such as age and prior admission for ACS or fetal hemoglobin measurements (HbF) on hemoglobin electrophoresis, may be interpreted with more confidence.

## 5. Conclusions

After 24 months of the pandemic, 185 SCD patients were infected with COVID-19 in the province of Quebec, Canada. Hospitalization and ICU admission rates were higher than in the general population of Quebec, but no deaths occurred. A history of ACS in the past year and systemic arterial hypertension were associated with a higher risk of hospitalization. Recruitment to the registry is ongoing to confirm the clinical risk factors and monitor the effect of new variants, repeat infections, and vaccinations on the impact of COVID-19 in SCD.

## Figures and Tables

**Table 1 jcm-11-07361-t001:** Characteristics of SCD patients with a diagnosis of SARS-CoV-2 infection between March 2020 and March 2022 in Québec.

	Patients *n* = 185
Median age (yr), (IQR)	12 [5–25]
0–9 yr, *n* (%)	74 (40)
10–19 yr, n (%)	45 (24)
20–29 yr, n (%)	34 (18)
30–39 yr, n (%)	17 (9)
40–49 yr, n (%)	7 (4)
50–59 yr, n (%)	5 (3)
60–69 yr, n (%)	3 (2)
Sex, male *n* (%)	94 (51)
SCD genotype, *n* (%)	
SS or Sβ^0^	121 (65)
SC or Sβ^+^	64 (35)
ABO blood group, *n* (%)	
A	36 (19)
B	31 (17)
AB	6 (3)
O	86 (46)
Unknown	26 (14)
Comorbidities *, *n* (%)	
Admission to ICU in the past 12 months	9 (5)
ACS in the past 12 months	14 (7)
Pulmonary hypertension	9 (5)
Hypertension	5 (3)
Chronic kidney disease	7 (4)
Obesity	22 (12)
SCD treatment *, *n* (%)	
Hydroxyurea	120 (65)
Chronic transfusion program	26 (14)
Crizanlizumab	1 (0.5)
None	48 (26)
Number of ED visits for SCD-related complications, *n* (%)	
0–3	173 (94)
4–6	12 (6)
7 or more	0
Median Hb F (IQR) **	15 [12–27]

* Patients may be on more than 1 treatment or have more than 1 comorbidity. ** For patients older than 1 year old and not on chronic exchange therapy *n* = 124.

**Table 2 jcm-11-07361-t002:** Clinical presentation and outcomes of SCD patients infected by COVID-19.

	Patients *n* = 185
Clinical presentation, *n* (%)	
Asymptomatic	30 (16)
COVID-19 classical symptoms ^1^	137 (74) *
Vaso-occlusive crisis	30 (16) *
Severity of infection, *n* (%)	
Mild (no hospitalization)	134 (72)
Moderate (hospitalization without ICU admission)	44 (24)
Severe (ICU admission)	7 (4)
Death	0

^1^ Classical symptoms of COVID-19 include cough, shortness of breath, fever, fatigue, myalgias, rhinorrhea, nasal congestion, anosmia and ageusia. * Twenty-one patients presented with concomitant COVID-19 symptoms and vaso-occlusive crisis.

**Table 3 jcm-11-07361-t003:** Complications and treatment of COVID-19 in hospitalized SCD patients.

	Patients *n* = 51
Complications during hospitalization, *n* (%)	
Acute chest syndrome	22 (43)
Pulmonary embolism or deep vein thrombosis	1 (2)
Acute kidney injury	3 (6)
Cardiac arrhythmia	1 (2)
Cardiac dysfunction	2 (4)
Oxygen therapy, *n* (%)	
Non-invasive oxygenation	10 (20)
Intubation and mechanical ventilation	6 (12)
Transfusion exchange therapy, *n* (%)	6 (12)
Other treatments received, *n* (%)	
Antibiotics	12 (24)
Therapeutic anticoagulation	3 (6)
Dexamethasone	2 (4)
Monoclonal antibody directed against IL-6	1 (2)
Monoclonal antibody directed against SARS-CoV-2 Spike protein	1 (2)

**Table 4 jcm-11-07361-t004:** Univariate analysis of potential predictors of hospitalization for SCD patients with SARS-CoV-2 infection. Relative risk ratios and 95% confidence intervals are presented.

	Hospitalization			
	No *n* = 134	Yes *n* = 51	Relative Risk (CI 95%)	
Median age, (IQR)	12 (4–25)	14 (5–25)		
0–23 months, *n* (%)	15 (11)	5 (10)	0.89 (0.40–2.00)	*p* = 0.78
2–9 yr, n (%)	42 (31)	14 (27)	0.87 (0.51–1.45)	*p* = 0.60
10–19 yr, n (%)	29 (22)	13 (25)	1.17 (0.69–1.97)	*p* = 0.58
20–39 yr, n (%)	38 (28)	14 (27)	0.97 (0.57–1.63)	*p* = 0.90
40–59 yr, n (%)	8 (6)	4 (8)	1.22 (0.53–2.83)	*p* = 0.64
≥60 yr, n (%)	2 (1)	1 (2)	1.21 (0.24–6.12)	*p* = 0.82
Sex, male *n* (%)	64 (48)	30 (59)	1.48 (0.59–2.23)	*p* = 0.17
Comorbidities, *n* (%)				
ICU admission in the past 12 months	5 (4)	4 (8)	1.66 (0.77–3.60)	*p* = 0.25
ACS in the past 12 months	6 (4)	8 (16)	2.27 (1.35–3.83)	*p* = 0.01
Pulmonary arterial hypertension	5 (4)	4 (8)	1.66 (0.77–3.60)	*p* = 0.25
Hypertension	1 (1)	4 (8)	3.06 (1.85–5.06)	*p* = 0.008
Chronic kidney disease	3 (2)	4 (8)	2.16 (1.09–4.30)	*p* = 0.07
Obesity	19 (14)	3 (6)	0.46 (0.16–1.36)	*p* = 0.11
SCD genotype, *n* (%)				
SS or Sβ^0^	84 (63)	37 (73)	1.40 (0.81–2.39)	*p* = 0.21
SC or Sβ^+^	50 (37)	14 (27)	0.71 (0.42–1.22)	*p* = 0.21
ABO blood group, *n* (%)				
A	23 (17)	13 (25)	1.41 (0.85–2.37)	*p* = 0.20
B	23 (17)	8 (16)	0.92 (0.48–1.77)	*p* = 0.81
AB	5 (4)	1 (2)	0.60 (0.09–3.62)	*p* = 0.54
O	62 (46)	24 (47)		
SCD treatment, *n* (%)				
Hydroxyurea	85 (63)	35 (68)	1.19 (0.71–1.97)	*p* = 0.51
Exchange transfusion program	16 (12)	10 (19)	1.49 (0.86–2.59)	*p* = 0.18
None	38 (28)	10 (19)	0.69 (0.38–1.28)	*p* = 0.23
Number of ED visits in the previous 12 months, *n* (%)				
No visit	80 (60)	33 (64)	1.17 (0.71–1.90)	*p* = 0.53
1–3	47 (35)	13 (26)	0.71 (0.41–1.24)	*p* = 0.21
4–6	7 (5)	5 (10)	1.56 (0.76–3.20)	*p* = 0.26
Median Hb (g/L) (IQR) ^1^	100 (89–108)	90 (77–98)	-	*p* = 0.04
Median % of Hb F (IQR) ^2^	17.8 (8–26)	12.0 (5–18)	-	*p* = 0.002
VOC at presentation of infection, *n* (%)	7 (5.2)	22 (43)	4.08 (2.77–6.01)	*p* ˂ 0.001

**^1^** Hb levels at the time of infection were available for 53 non hospitalized patients and all hospitalized patients. **^2.^** Patients under the age of one year old or on regular exchange transfusion therapy were excluded from this analysis (non-hospitalized *n* = 92, hospitalized *n* = 32).

**Table 5 jcm-11-07361-t005:** Rates of infection, hospitalization, and ICU admission compared to the general population of Quebec, Canada.

	SCD Population *n* = 1500	Quebec Population *n* = 8,639,742	Relative Risk (CI 95%)	
Infection prevalence	103/1500	455,527/8,639,742	1.30 (1.08–1.56)	*p* = 0.005
Hospitalization rate	31/103	26,628/455,527	5.15 (3.84–6.91)	*p* ˂ 0.001
0–9 yr	9/32	271/48,824	50.67 (28.78–89.28)	*p* ˂ 0.001
10–19 yr	5/19	179/60,346	88.72 (41.22–190.90)	*p* ˂ 0.001
20–29 yr	8/27	793/70,465	26.33 (14.66–47.28)	*p* ˂ 0.001
30–39 yr	4/11	1398/66,212	17.22 (7.87–37.7)	*p* ˂ 0.001
40–49 yr	1/6	1865/66,495	5.94 (1.01–35.58)	*p* = 0.04
50–59 yr	3/5	3220/55,145	10.28 (5.02–21.03)	*p* ˂ 0.001
60–69 yr	1/3	4217/33,892	2.00 (0.37–10.97)	*p* = 0.45
ICU admission rate	6/103	5499/455,527	4.56 (2.09–9.93)	*p* ˂ 0.001

**Table 6 jcm-11-07361-t006:** Comparison of the Quebec SCD—COVID-19 population to the French, United States of America, and SECURE-SCD registries.

	Quebec Registry	French Registry Arlet et al. [10]	United States of America Registry ^1^ Singh et al. [11]	International SECURE-SCD Registry [12]
Number of patients	185	319, all hospitalized	312	1045
Period of observation	24 months [March 2020–April 2022]	14 months [March 2020–May 2021]	8 months [January 2020–September 2020]	Updated March 2022
Age, yr	12 (median)	26 (median)	31 (mean)	20 (mean)
Sex, male %	51	49	37	48
Comorbidities, %				
Previous ACS	7 (past year)	57 (anytime)	-	29 (past 3 years)
Hypertension	3	9	27	12 (among >18 yr)
Chronic kidney disease	4	-	-	5
Diabetes	0	2	35	6 (among >18 yr)
Obesity	12	-	18	4 (among <18 yr)
Pulmonary hypertension	5	-	-	6
SCD Genotype, %				
SS	63	87	-	64
SC	28	10		23
SCD treatment, %				
Hydroxyurea	65	56	-	55
Exchange transfusion program	14	13		11
None	26	-		-
Severity of infection, %				
Mild (no hospitalization)	72	-	81	~60
Moderate (hospitalization without ICU admission)	24		19 (all admission)	~38
Severe (ICU admission)	4			8
Risk factors for hospitalization or death	Previous ACS in the past year, arterial hypertension	SC genotype associated with mechanical ventilation and death	-	>2 ED visits for pain and previous ACS in the past 3 years, Pulmonary hypertension [13]
Death, %	0	2.2	3.2	1.8

^1^ We only report data regarding SCD population. Individuals with sickle trait were not included in this table.

## Data Availability

The data presented in this study are available on request to the corresponding authors. All patient data were acquired from our institutional data base as per our ethical and research committee approval. No data was shared with any third party or any public data sets.

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
