# Peer review of "COVID-19 and Sickle Cell Disease in the Province of Quebec, Canada: Outcomes after Two Years of the Pandemic"

_jcm, 2022, doi:10.3390/jcm11247361_

Round 1

Reviewer 1 Report

Sickle cell disease patients are considered at high risk when infected by SARS-CoV-2 because of impaired immunity, systemic vasculopathy that contributes to multiple organ dysfunction, and high risk of thrombosis. They have pre-existing chronic morbidities and they are frequently splenectomised or asplenic increasing the risk of bacterial infection but we don’t know the risk of severe viral illness.

In this context, with your results and those of the literature, could you put into perspective the advanced mechanisms of high risk of morbidity/mortality of sickle cell patients infected with SARS-CoV-2?

In the USA study, death was more often observed in males. Did you also observed gender differences?

If feasible, it would be interesting to provide a table with your data and those of the literature for an easier comparison and which would support your discussion.

Were you able to pick up the Ct values for the SARS-CoV-2 RT-PCRs? If so, is there a correlation with the symptoms observed? Or have you observed different results during different waves of infection?

Author Response

Dear Reviewer, 

Thank you very much for reading and commenting our manuscript. 

Please find attached a detailed answer, under reviewer 1.

Thank you.

Reviewer 2 Report

Very interesting and well written work.

Authors could add two references from Greece showing similar results.

1.P116 THE IMPACT OF COVID19 PANDEMIC ON SICKLE CELL
MANAGEMENT: EXPERIENCE OF A SINGLE PEDIATRIC CENTER
Maria Vousvouki, ; Aikaterini Teli,; Alkistis Adramerina,
Stamatia Theodoridou,  Marina Economou, Hemasphere 2022;6:S1.

2.Hemoglobinopathiew and Covid -19: The experience of a center in Northern Greece,

 Christos Varelas et alHemoglobin vol 46, 2022, Issue 2.

Author Response

Dear Reviewer, 

Thank you very much for reading and commenting our manuscript. 

Please find attached a detailed answer, under reviewer 2.

Thank you.

Reviewer 3 Report

Castonguay et al report the number and characteristics of SCD patients with confirmed SARS-41 CoV2 infection recorded from March 2020 to April 2022 in the province of Quebec, Canada, and the risk factors  of hospitalization and admission to intensive care unit

Few comments:

-        One of the major limitation of the study is the biais in the patients recruitment. As I understand it (and this should be more clearly stated in the method section), patients were tested with COVID when they went to the hospital for an overt symptom (infection  or SCD-related symptom).

In this way, all patients with only few, or no symptoms are missing. Thus, when the authors say several time in the paper that “After 24 months of the pandemic, 185 SCD patients were infected with SARS-COV2  in the province of Quebec, Canada”, is it realistic? Perhaps the conclusions should be moderated by taking into account possible asymptomatic patients. Wouldn't it be more acurate to say that “185 patients with SCD who visited the ED were infected with SARS-COV2 ?”.

I will be even more cautious as there is probably a bias among patients who consulted the ER for a symptom: It is likely that patients without comorbidity were less likely to come to the hospital in case of symptoms and therefore to be screened, while patients who were already the most severe, as well as young children, had to consult more systematically.

-        In the Table 2: 16% patients were reported to be asymptomatic. What were the reasons for the test for them? Once again this precise criteria of inclusion should be clarified in the beginning.

-        The table 3 present the Complications and treatment of COVID-19 in hospitalized SCD patients: I think that the age/Genotype and frequences  Co-morbidities of the patients should also be precised.

-        Table 4: the groups of age are not divided in the same way as in table 1. Specially for the Children, grouping the patients from 2 to 19 seems a little bit large considering the specificity of pediatric populations and the importance of pediatric patients in this cohort . 3 groups (0-2; 2-10; 11-19 ) would be more appropriated.

-        Table 5: why is the infection prevalence 103/1500 while the results reported 185 cases of infection?

-        SC/Sbeta+ represent around a third of the patients with Sars-Cov2. What is the proportion of SC patients in the whole cohort? Currently the proportion of SC patients in European and North American cohort is around 20%, so I wonder if proportionally they were an over-representation in the SC patients in the symptomatics patients?

-        Why HbF is the only biological parameters reported? What about the Hb level for exemple?

-        Within the discussion, the authors hypothesize that the universal health care access in Quebec may partially explain the difference between some of their results and the reported North-American, international and French Cohorts. If I’m not mistaken, french patients with SCD are followed in public and free-access specialized SCD center and many patients of the SECURE SCD report were also followed in countries in Europe who benefit of a free and funded system of health?

Author Response

Dear Reviewer, 

Thank you very much for reading and commenting our manuscript. 

Please find attached a detailed answer, under reviewer 3.

Thank you.

Round 2

Reviewer 3 Report

The article has been substantially revised.

Nevertheless, I continue to disagree with the conclusion that "After 24 months of the pandemic, 185 patients with SCD were infected with SARS-COV2 in the province of Quebec, Canada," since the majority of patients with few or no symptoms, and therefore did not require PCR testing or an emergency room visit, are missing from the study.